# MapLearn: Indoor Mapping using Audio

## Abstract

Cameras and LIDARs are established methods to generate the map (or floorplan) of an indoor environment. This paper investigates the feasibility of using audio to learn the map. We aim to transmit audio beacons from a mobile device (say a smartphone) and record its reflections from the environment. Assuming known user locations, and recordings from multiple locations along walked paths, we aim to learn the 2D floorplan of the area. We use a conditional GAN (cGAN) architecture but prevent it from over-fitting using knowledge of indoor signal propagation. We pre-train our model on simulated data – thousands of high-fidelity audio measurements on hundreds of synthetic floor plans – and then test on 4 real environments in our home and office buildings. Results show that the generated maps are fairly accurate (in terms of precision and recall) even though no training was performed in real rooms. We have assumed clutter-free rooms; coping with clutter remains a topic for continued research.

## 1 Introduction

Indoor localization and mapping are important building blocks to a range of spatial applications, including navigation, augmented reality, digital twins, and context-aware voice assistants (like Alexa) Thrun et al. (2002); Hess et al. (2016); Shen et al. (2020); Schönberger et al. (2018). Decades of research has focused on localization but relatively less effort has been directed to mapping. Today's mapping solutions use cameras and LIDARs to scan the environment Zhang & Singh (2014); Schonberger & Frahm (2016), and while this generates detailed and accurate maps, they incur privacy concerns, especially in homes Fraccaro et al. (2020). Moreover, the detailed maps from cameras/LIDARs may be an overkill for many applications; a simple floorplan contour may suffice in most cases.

This paper aims to learn the floorplan of an environment by transmitting audio beacons from a mobile device (e.g., smartphone) and recording its reflections at the same device. The audio signal radiates in all directions, bounces off walls and objects, and echo back to the device, offering information about the surrounding geometry. We intend to gather these measurements from multiple locations as the user walks around in her home. Assuming user locations (to the extent feasible from today's localization technology), we aim to learn the 2D map (or floorplan) of the place.

At the heart of our approach is a combination of signal processing and learning. Briefly, we first estimate the room impulse response (RIR) Habets (2006) and then make spatial inferences on the RIR to outline open spaces. Walls and reflective objects can only lie outside these open spaces. We now train a conditional generative adversarial network (cGAN) Mirza & Osindero (2014) that learns to erect walls around these open spaces in a manner that matches the audio measurements. Our model, `MapLearn`, copes with challenges such as rotation ambiguities and room symmetry. The final outcome is a 2D floorplan, evaluated by calculating 2D precision and recall. Results show that even when a user walks through $60\%$ of the grids in a $20m \times 20m$ home, the median wall error is less than 60cm. Visually, the estimated floorplan bears good resemblance to the ground-truth.

Figure 1 shows snapshots from the overall `MapLearn` pipeline. Figure 1(a) shows an example synthetic room in which audio signal propagation is simulated; the room walls are coated with realistic materials so the audio behavior models reality. Figure 1(b) shows an example trajectory on which the user walks. Once training is complete, Figure 1(c) shows an example real environment in which measurements are made from a speaker-microphone pair. Figure 1(d) and (e) show the comparison between the true and estimated floorplans. While `MapLearn`'s results are robust and fairly promising, there is room for improvement in mainly two directions. (1) Our environments include minimal furniture and clutter. Coping with reflections from clutter is a next step to this paper. (2) Our solution degrades once users walks through less than $60\%$ of the grid cells in the overall floorplan. We believe there is room to learn more effectively even from sparser measurements of the environment.

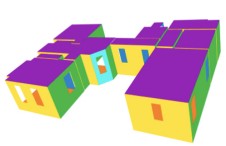 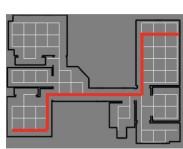 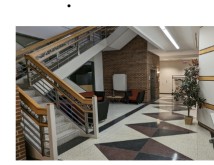 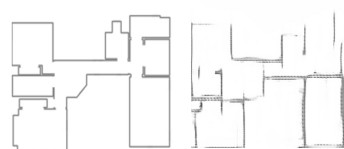

Figure 1: (a) 3D mesh of simulation environment, (b) an example trajectory, (c) real world experimental environment, (d) groundtruth floorplan, (e) estimation result.

In sum, the contributions of `MapLearn` are:

1. Learning the map of an indoor environment by guiding a conditional GAN with spatial hints derived from signal processing.

2. A practical solution that only uses audio beacons from a single speaker and microphone pair, and allowing the user to walk around in random trajectories in the home.

3. Robust performance through simulated training that generalizes to real (clutter-free) homes.

Prior work closest to `MapLearn` is BatVision Christensen et al. (2020). Authors use audio reflections recorded by two microphones to infer the spatial scene (including depth). However, the method targets imaging local scenes in front of the device's current location. In contrast, `MapLearn` combines mobility and environmental reflections to stitch together the entire 2D floorplan of a home. Inferring this global view with a single microphone, and from incomplete coverage of the home, makes the problem unexplored. Section 5 discusses further on the difference with SLAM, synthetic radars, and other imaging techniques in literature.

## 2 METHOD

The input to `MapLearn` is a set of user locations, $L$, the audio signal recorded from all these locations, $R$, and the waveform of the transmitted audio beacon, $s_0(t)$ (the same waveform is transmitted from all locations). `MapLearn`'s output is a map (or floorplan), $y$, of the indoor environment.

### 2.1 PROBLEM FORMULATION

We cast the floorplan estimation problem as learning a function $G$ that transforms the sensor input $x = \langle R, L \rangle$ into floor plan $y$, i.e., $y = G(x)$. We use the conditional generative adversarial network (cGAN) Mirza & Osindero (2014) as the backbone architecture for `MapLearn`. The conditional GAN takes a sample $x$ from the sensor input distribution and outputs a prediction $y$ in the floor plan's distribution. During the process, the discriminator learns to identify when estimated floorplans are implausible (i.e., low likelihood given the true floorplans), and the generator attempts to fool the discriminator by minimizing the "divergence" between estimated floorplans and the true ones. Analytically, the network plays the classic mini-max game, ultimately learning to maximize $p(y|x)$.

$$\underset{G}{\text{argmin}} \ \underset{D}{\text{argmax}} \ E_y\big[logD(y|x))\big] + E_x\big[log(1 - D(G(x)|y)\big] \tag{1}$$

Here $D$ and $G$ are the discriminator and generator, respectively. Once training is complete, the floor plan $y$ is obtained from applying the generator, $G(x)$. We attempted to train an expressive cGAN model based on equation 1 to verify if the floorplan estimation can be achieved. Unfortunately, given the highly inverse nature of the problem (i.e., from received signal to indoor map), and the intrinsic data hungry nature of cGANs, convergence was difficult; results were poor.

Based on principles of signal propagation in reflective environments, we recognize opportunities to guide the cGAN with spatial hints. This motivates what we term as a `hint-map` $H(x)$ as an augmentation to cGAN. Thus instead of directly training on the objective function in Equation 1, we modify the training objective to be

$$\underset{G}{\text{argmin}} \ \underset{D}{\text{argmax}} \ E_y\big[logD(y|x, H(x))\big] + E_x\big[log\big(1 - D\big(G(x, H(x))|y\big)\big)\big] \tag{2}$$

The following sections presents the details on $H(x)$ and it's introduction into the architecture.

## 2.2 HINT MAP $H(x)$ GENERATION

Consider an audio beacon transmitted and received from any location $l_i$ in a room. The beacon signal $s_0$ bounces on surrounding walls and returns to the microphone to produce a recording, $r$. The received signal $r$ is a time convolution between $s_0$ and the room acoustic impulse response $h$. Each echo received at the microphone is caused by a reflector $w$ producing a $\tau(w)$-delayed and $A(w)$-attenuated copy of the original source signal $s_0$. Equation 3 models the process ($*$ is time-domain convolution):

$$h = \sum_{w \in \Psi} A(w)\delta(t - \tau(w)) \qquad r = s_0 * h = s_0 * \sum_{w \in \Psi} A(w)\delta(t - \tau(w)) \qquad (3)$$

To extract acoustic channel $h$, we perform a time-domain deconvolution between the received signal $r$ and the source signal $s_0$. Figure 2 (a) visualizes the acoustic channel $h$, composed of early reflections from nearby objects and late reverberations from higher-order reflections. We intend to derive *robust geometric hints* from $h$. Obviously, each peak with delay $\tau(w)$ corresponds to a reflecting path with a total path length of $\tau(w) \cdot v$, where $v$ is the speed of sound. If this path is due to a *single* reflection, then the corresponding reflector should exactly lie on the circle centered at the measurement location $l_i$, with radius

$$\rho_i = \tau(w) \cdot v/2 \qquad (4)$$

We spatially encode the acoustic channel $h$ to the 2D floor plan based off Equation 4. Figure 2(b) illustrates this visually where each white circle corresponds to an echo; the radius of the circle models the distance traversed by that echo. Observe the nearest wall exactly lies on the smallest circle corresponding to the first peak in $h$. We know there will be no obstacles within this circle $C(l_i, \rho_i)$ because any reflector within this circle should have created an earlier echo with smaller delay.

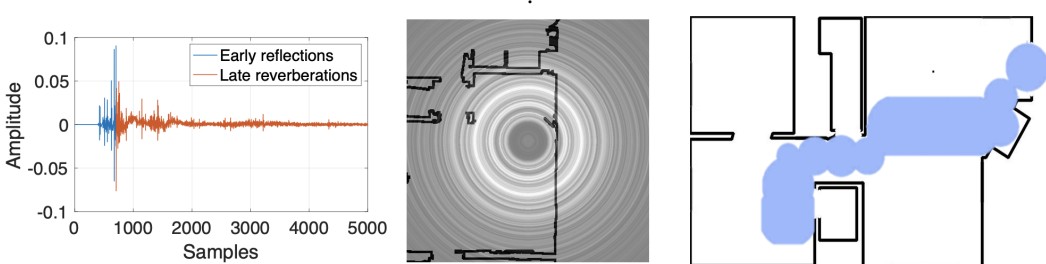

Figure 2: (a) Deconvolved acoustic channel $h$. (b) Spatially encoding the acoustic channel $h$ to the 2D floor plan. The closest obstacle exactly corresponds to the first peak in early reflection. (c) Union of the first peak circle for all measurement locations along a certain trajectory forms the hint-map.

As users walk along a random trajectory, each measurement from a distinct location produces a circle $C$. We define the hint map $H(x)$ as the union of all the first peak circles along the trajectory:

$$H(x) = \cup_{i=1}^{n} C(l_i, \rho_i) \qquad (5)$$

Figure 2 (c) plots the hint map $H(x)$ for a trajectory. We believe $H(x)$ is robust across various settings and use it as a reliable guidance for training the cGAN.

## 2.3 GENERATOR DESIGN

The generator $G(x)$ is a two step process. In (step 1), we learn the room structure based on local audio measurements; in (step 2), we stitch the rooms together, and combine with hint map $H(x)$ to globally optimize for the final floorplan. This design is motivated from the weak audio penetration through walls; since most of the signal remains inside a room (as opposed to penetrating and reflecting back from adjacent rooms), we leverage the opportunity to learn each room separately and then stitch them appropriately. This extends robustness since we can only build off strong reflections (as

opposed to weak reflections returning from other rooms). We sacrifice some ability to understand the adjacency of rooms, but we rely on the mini-max game to learn that (relatively easier) pattern.

**Step 1: Individual room estimation.** Given the noisy nature of the room's impulse response (RIR), directly learning on RIR is highly prone to over-fitting. To alleviate this, we again turn to signal propagation models. As shown in figure 2, the acoustic RIR $h$ is composed of a few early reflections followed by a long tail of reverberation Bradley et al. (2003). The reverberation decay is closely related to the shape of the room, while timings of the early peaks are a function of the user's location inside the room. Thus, instead of inputting RIR $h$ into the network, we input the envelop $E(h)$ which is far less noisy. Figure 3(a) shows both $h$ and the envelop $E(h)$. The first few peaks of $h$ have been preserved well by the envelope whereas the later (less important) peaks are only captured through a decaying tail. Hence, $E(h)$ preserves the salient spatial information necessary to learn the low resolution contours of the environment; this reduces over-fitting and aids in robustness and generalization to real-world environments.

Next, we fit a rectangle to each room and train a MLP network that regresses the corners of the room from the envelop $E(h)$. At this time, we assume the measurement location as origin $< 0, 0 >$ in the room's local reference frame (we will transform all predictions to the global frame in step 2). Figure 3(b) shows a measurement location in red and the true room around it; Figure 3(c) plots the corresponding MLP-predicted room with the red point as origin.

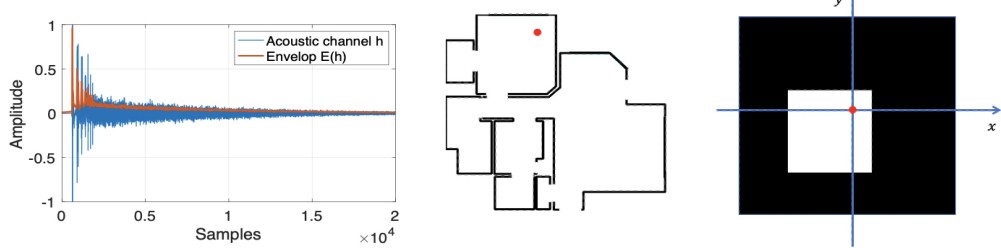

Figure 3: (a) Raw RIR $h$ and the envelope $E(h)$ capture spatial features of the room. (b) Groundtruth room with measurement from the red location. (c) The MLP-predicted room from the red location.

Due to the omni-directional gain patterns of speakers and microphones, the RIR $h$ is rotation invariant, i.e., all rotations of the same room, around the measurement location $l_i$, will produce the same RIR $h$. Figure 4(a) helps understand this invariance. With 2 measurements however, the rotation invariance is mitigated, however, 2 rooms are still possible that produce the given $h$. These 2 rooms are the reflections along the straight line joining the 2 measurement points (see Figure 4(b)). To resolve all ambiguity, we input 4 nearby measurements into the MLP encoder, as shown in Figure 4(c). While 3 measurements are adequate, we use 4 measurements to over-determine the system for better reliability and easier integration into our grid-based pipeline.

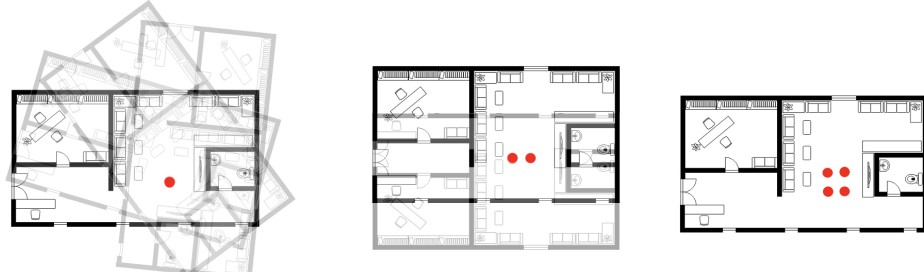

Figure 4: Ambiguity exists in acoustic measurement due to the omni-directional nature of our sensor. (a) rotational ambiguity for a single measurement. (b) flipping ambiguity for two measurements. (c) we use four measurements for one estimation to resolve ambiguity.

**Step 2: Stitching rooms into a floorplan.** Given we know the user's location/trajectory in an absolute reference frame, we now shift each room to be around that absolute location. We use a UNet architecture Ronneberger et al. (2015) to fuse the shifted rooms (in the global frame) and the hint map. Together they should help learn the final floorplan.

## 2.4 NEURAL NETWORK IMPLEMENTATION

Figure 5 shows `MapLearn`'s complete architecture. Users walk along random trajectories – audio recordings from these locations are input to the generator. For generator step 1, we use a MSE loss on the corner coordinates of the room. We use a patch GAN discriminator to distinguish the real floor plan from the generated floor plan with the conditional inputs. Inspired by Isola et al. (2017), when training step 2 of the generator, we use a cGAN loss and a L1 loss on the generated floor plan to reduce blurring effects. The final objective function is:

$$\underset{G}{\text{argmin}}\, \underset{D}{\text{argmax}}\, \mathcal{L}_{cGAN}(G, D) + \lambda ||G(x, H(x)) - y||_1 \tag{6}$$

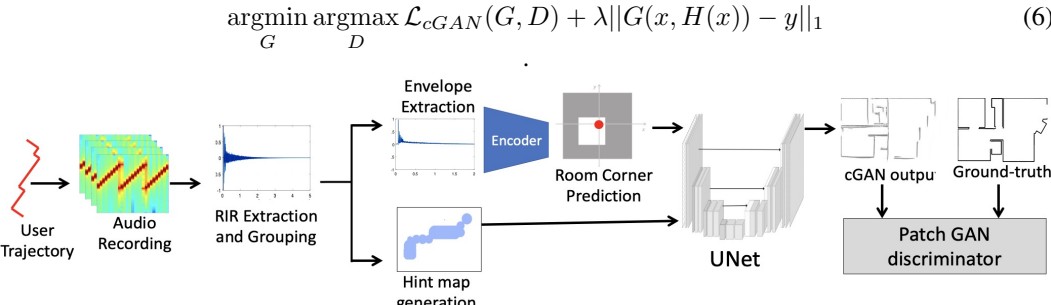

Figure 5: `MapLearn` architecture: locations and RIRs are input to cGAN, along with hint maps. The generator is a two step process composed of (1) per room estimation and (2) global optimization.

## 3 DATASET CREATION

**Simulation.** Due to the need for substantive training data, and the challenge in collecting data from real environments, we aim to train on simulation data. This approach is gaining popularity and proving successful in massive training-data generation. We use the simulation engine from SoundSpace 2 Chen et al. (2020; 2022) to generate sound ray-tracing on Zillow's indoor dataset (Zind) Cruz et al. (2021). The Zind dataset includes 1000+ RGB scans of real floor plans. For each scan, we generate the 3D mesh of the environment using the tool associated to Zind. We added ceilings to the 3D reconstruction and scaled the dataset to ensure that the rooms have realistic sizes. We created $1m \times 1m$ grids in all the environments and performed audio simulation on the grid points, i.e., by inserting speakers and probe microphones co-located at each measurement location. We then generated a number of random trajectories (with random start and end grid points). Figure 6 plots samples from our simulation pipeline.

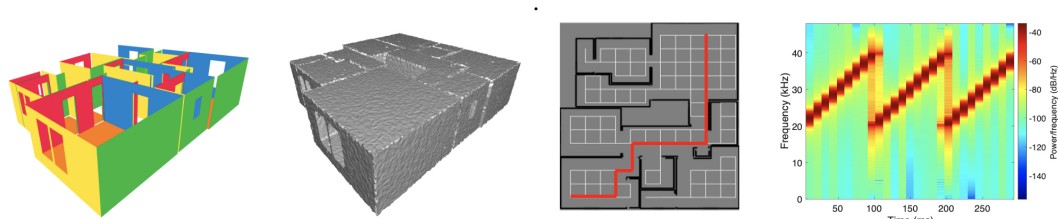

Figure 6: Simulation pipeline: (a) raw 3D mesh from Zind, (b) added roughness and ceiling to the meshes, (c) navigation graph and trajectory generation, (d) acoustic simulation along the trajectories

To model scattering of audio signals after reflections, we add white Gaussian noises to the walls, ceilings, and floors of the environment – this introduces roughness in the 3D mesh. Figure 7 shows a comparison between (a) the real RIR from a real $3m \times 3m$ empty room, (b) the RIR simulated with the off-the-shelf ZIND dataset, and (c) the RIR generated after our updates. Our simulated RIR bears good similarity to the real channel model. We simulate a total of $> 40k$ audio measurements at a sample rate of $96kHz$.

**Real indoor data.** For test data, we performed real-world measurements using ultrasound beacons (20 to 40 $kHz$) in a university building and two home environments. This data set contains $\approx 200$ meters of trajectories, with $> 10k$ RIRs; the audio data is sampled at $96kHz$ and the audio beacons are scheduled at 10Hz. For user localization, we also collect the IMU sensor data from the user's

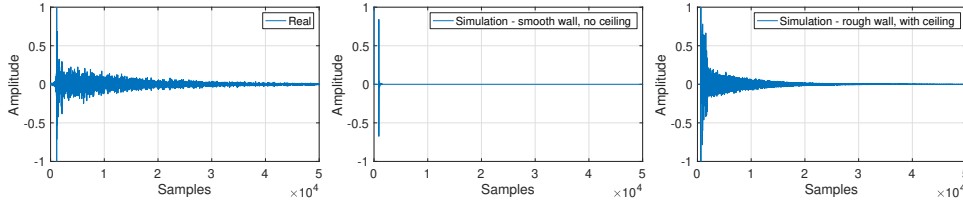

Figure 7: Comparison between (a) acoustic channel measured in real room. (b) Simulation result when we directly run SoundSpace on Zind mesh. (c) After our updates to the simulator.

Samsung Galaxy S21 smartphone, at a sampling rate of $100Hz$. We implemented an existing IMU-based localization method Yang et al. (2020) – the code is available. The ground-truth floor plans were collected from the building administrators. Figure 8 shows sample indoor environments: the first one from simulations and the remaining from university buildings and residential apartments.

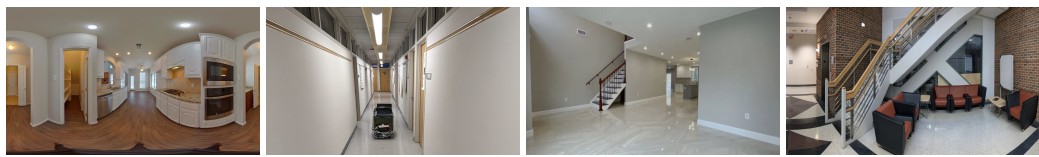

Figure 8: Sample environments for evaluation; (a) is a simulated environment from the ZIND dataset while (b) - (d) are real environments from homes and offices.

## 4 EVALUATION

**Evaluation metric.** To compare the estimated map against the ground-truth, we define a metric similar to precision and recall. Note that any wall is a contiguous sequence of pixels (Figure 9 shows true walls in black and estimated walls in red). Our main idea is to expand the width of all true walls by $d$ cms and count the fraction of estimated pixels that fall inside the wider true walls. We call this `precision(d)`. Figure 9(a) visualizes the idea for only a single wall, where `precision(d=20cm)` is 0.8 since $80\%$ of the estimated (red) wall are within 20cm expansion of the true (black) wall.

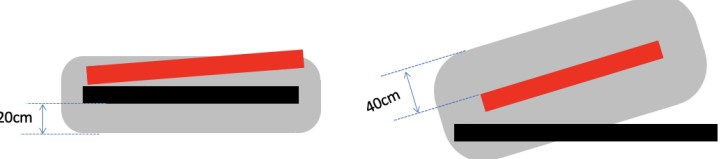

Figure 9: Visualizing evaluation metrics: `precision(d=20cm)` and `recall(d=40cm)`

We also define `recall(d)` where the estimated red walls are expanded by $d$ cms, and we compute the fraction of true black pixels that fall within these wider red walls. Figure 9(b) shows an example where `recall(d=40cm)` is 0.5. We report `precision(d)` and `recall(d)` for $d = 20, 30, 40, 60,$ and 80cm.

**Baseline / Ablation / Sensitivity.** Prior work has reported ranging errors for individual walls but we were not able to find a clear baseline for the whole map estimation. Hence, we report `precision(d)` and `recall(d)` against the ground-truth, but perform ablation studies with only the hint map as an input to the cGAN (i.e., eliminating the contribution of the RIR). We also report sensitivity of the performance to varying amount of coverage, i.e., the fraction of the floorplan area that the user walks for audio measurements. We also show the performance sensitivity to user localization error, given that a real system will not know the accurate user location along the trajectories.

### 4.1 MAPLEARN EVALUATION: SIMULATION RESULTS

We evaluate the performance of `MapLearn` on 16 unseen floor plans, each drawn from real floorplan datasets and modeled through detailed simulation. Table 1 reports the results com-

paring `MapLearn` with the hint-map only based solution. Evidently, `MapLearn` achieves `precision(d=40cm)` of 0.567 implying that 56.7% of estimated walls are within $40cm$ of the true walls in the actual floorplan. The `recall(d=40cm)` is also similar: 0.505. The hint-map based solution performs worse, achieving `precision(d=40cm)` of 0.471. This captures the contribution of room reverberations that help estimate corners and other protruding partitions in the environment. Overall, `MapLearn` offers a 5% gain over only using the hint-map.

| d(cm) | 20 | | 30 | | 40 | | 60 | | 80 | |
|---|---|---|---|---|---|---|---|---|---|---|
| Prec./Rec. | $P(d)$ | $R(d)$ | $P(d)$ | $R(d)$ | $P(d)$ | $R(d)$ | $P(d)$ | $R(d)$ | $P(d)$ | $R(d)$ |
| MapLearn | 0.365 | 0.336 | 0.457 | 0.398 | 0.567 | 0.505 | 0.659 | 0.588 | 0.761 | 0.675 |
| Hint Map | 0.312 | 0.377 | 0.372 | 0.439 | 0.471 | 0.539 | 0.553 | 0.622 | 0.666 | 0.705 |

Table 1: Quantitative results for MapLearn and hint map only comparison on simulation data for precision $P(d)$ and recall $R(d)$, with different error margins $d$.

Figure 10 presents qualitative visualization of 6 different samples estimated by `MapLearn`. The top row shows the original 3D mesh followed by the ground-truth 2D floorplan in the next row. The third row shows the hint-map only solution, while the fourth row is the output from `MapLearn`. `MapLearn` and hint-map, both tend to infer the contour of the floorplan well, however, `MapLearn` learns the corners and finer structures better. The actual positions of the walls are also learnt better in `MapLearn` since the reverberations embed spatial information about the environment through it's amplitude and time-differences between successive echoes.

.

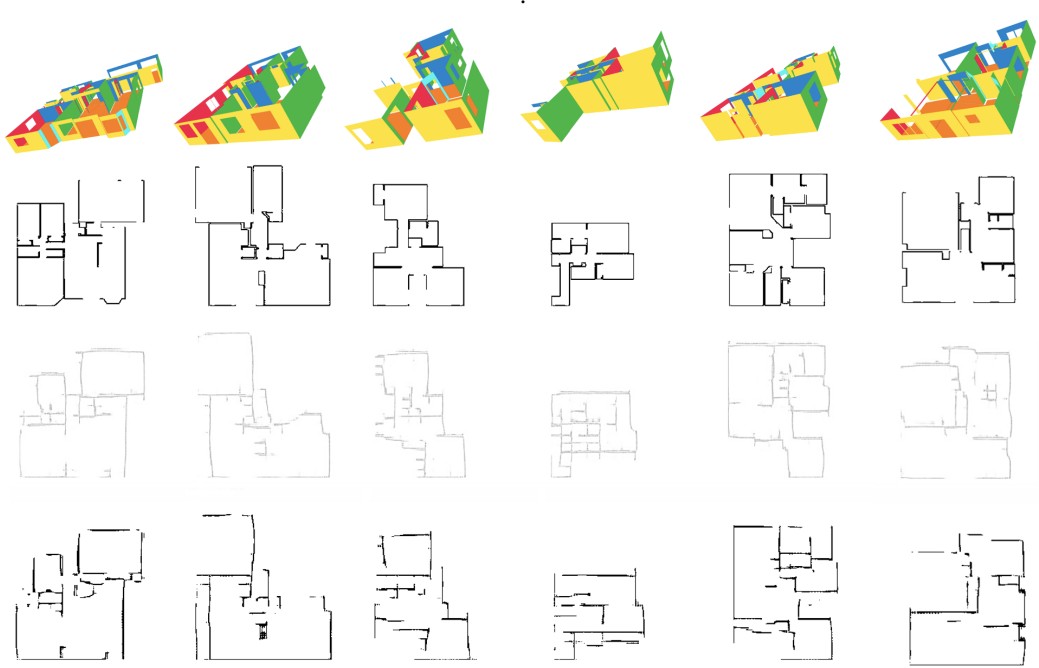

Figure 10: Qualitative results for `MapLearn` on simulation data. Row (a) 3D mesh of experimental environment, (b) floor plan ground-truth, (c) floor plan estimation for ablation study, hint map only, (d) floor plan estimation for `MapLearn`.

**Sensitivity to coverage.** `MapLearn`'s performance will depend on the amount of measurements available from different parts of the indoor environment. Assuming the whole floorplan is divided into grid cells, we define `coverage` as the percentage of grid cells from which audio measurements were made. Table 2 shows `MapLearn`'s performance for varying coverage; naturally, the performance degrades when coverage is smaller. The degradation is reasonably graceful because the reverberations still offer information about the wall locations and sizes of rooms even if the user is not walking near them. Of course, the performance below 70% is unimpressive and part of the reason is that audio signals were transmitted at low power in view of practical usage. Speakers in mobile devices like smartphones have limited power; this low power attenuates the third and fourth

order reflections substantively, allowing little geometric information to be preserved in the returning echoes, especially when the user is walking far away from the walls.

| d(cm) | 20 | | 30 | | 40 | | 60 | | 80 | |
|---|---|---|---|---|---|---|---|---|---|---|
| Prec./Rec. | $P(d)$ | $R(d)$ | $P(d)$ | $R(d)$ | $P(d)$ | $R(d)$ | $P(d)$ | $R(d)$ | $P(d)$ | $R(d)$ |
| $Cov = 50\%$ | 0.232 | 0.268 | 0.276 | 0.322 | 0.353 | 0.407 | 0.415 | 0.469 | 0.491 | 0.544 |
| $Cov = 60\%$ | 0.245 | 0.299 | 0.289 | 0.357 | 0.376 | 0.451 | 0.451 | 0.522 | 0.528 | 0.606 |
| $Cov = 70\%$ | 0.254 | 0.330 | 0.303 | 0.392 | 0.389 | 0.491 | 0.461 | 0.566 | 0.538 | 0.649 |
| $Cov = 80\%$ | 0.272 | 0.353 | 0.320 | 0.425 | 0.427 | 0.531 | 0.520 | 0.610 | 0.610 | 0.697 |
| $Cov = 90\%$ | 0.335 | 0.393 | 0.397 | 0.468 | 0.517 | 0.577 | 0.612 | 0.653 | 0.710 | 0.738 |

Table 2: Effect of decreasing coverage $Cov$ on `MapLearn`'s precision $P(d)$ and recall $R(d)$.

**Sensitivity to user location error.** Simulations so far assume accurate measurement locations. In reality, indoor localization methods will incur errors (as will be the case in our real-world experiments). To assess sensitivity to localization error, we inject random Gaussian noise around the user's location, $l_i$, i.e., the 2D noise vector $n$ is sampled as $\mathcal{N}(n; l_i, \sigma^2 I)$. Table 3 reports the effect of varying $\sigma$ at $85\%$ coverage. `MapLearn`'s performance degrades especially when $\sigma = 40cm$ or more. However, even with $\sigma = 80cm$, the precision and recall are still around $50\%$, meaning that half of the walls are correctly positioned within $d = 80cm$ from the true walls. Given floorplans are around $20m \times 20m$ in area, we believe this level of mapping error is tolerable to many applications.

| d(cm) | 20 | | 30 | | 40 | | 60 | | 80 | |
|---|---|---|---|---|---|---|---|---|---|---|
| Prec./Rec. | $P(d)$ | $R(d)$ | $P(d)$ | $R(d)$ | $P(d)$ | $R(d)$ | $P(d)$ | $R(d)$ | $P(d)$ | $R(d)$ |
| $\sigma = 20cm$ | 0.355 | 0.318 | 0.428 | 0.380 | 0.551 | 0.480 | 0.632 | 0.561 | 0.723 | 0.654 |
| $\sigma = 40cm$ | 0.325 | 0.271 | 0.385 | 0.322 | 0.487 | 0.401 | 0.565 | 0.462 | 0.657 | 0.549 |
| $\sigma = 60cm$ | 0.304 | 0.239 | 0.355 | 0.282 | 0.447 | 0.349 | 0.523 | 0.400 | 0.606 | 0.472 |
| $\sigma = 80cm$ | 0.284 | 0.245 | 0.341 | 0.285 | 0.420 | 0.352 | 0.487 | 0.405 | 0.581 | 0.473 |

Table 3: Effect of localization error (modeled by Gaussian noise parameter $\sigma$) on $P(d)$ and $R(d)$.

## 4.2 MAPLEARN EVALUATION: REAL-WORLD EXPERIMENTS

We evaluate `MapLearn` in the real world by transmitting audio beacons from an omnidirectional speaker and simultaneously recording using a microphone. We pre-process the recorded audio data to remove the direct signal from the speaker to the microphone; the residue signal contains the reflections from walls and is fed as input to `MapLearn`. Observe that `MapLearn` is trained on simulated data, and even though we modeled walls with roughness of materials, there is still an understandable gap between simulation and reality. Moreover, simulated rooms were free of furniture, while real environments unavoidably had sparse furniture. We also implemented an indoor localization method from a past work; this method also incurred error in tracking the user's motion.

Figure 11 shows the final maps learnt by both `MapLearn` and hint-map. The measurements were made from 10 minutes of slow walking in indoor areas of around $15m \times 15m$. The precision and recall values are reported in Table 4 and align with the outcomes of simulations. This is promising (and somewhat surprising) in how the simulated training generalizes to the real world despite various sources of errors. Of course, with increasing clutter the reverberations are far more complex and the results degrade sharply. We leave the treatment with furniture for future work.

| d(cm) | 20 | | 30 | | 40 | | 60 | | 80 | |
|---|---|---|---|---|---|---|---|---|---|---|
| Prec./Rec. | $P(d)$ | $R(d)$ | $P(d)$ | $R(d)$ | $P(d)$ | $R(d)$ | $P(d)$ | $R(d)$ | $P(d)$ | $R(d)$ |
| MapLearn | 0.197 | 0.196 | 0.236 | 0.249 | 0.325 | 0.341 | 0.421 | 0.435 | 0.581 | 0.559 |
| Hint Map | 0.179 | 0.151 | 0.223 | 0.192 | 0.336 | 0.276 | 0.429 | 0.352 | 0.525 | 0.455 |

Table 4: Quantitative results for MapLearn and hint map only comparison on real world data for precision $P(d)$ and recall $R(d)$, with different error margins $d$.

## 5 RELATED WORK

■ **Image sensors:** The vast majority of indoor mapping methods are built on imaging sensors, such as cameras, LIDARs, etc. Liu et al. (2015); Ito et al. (2014); Cruz et al. (2021); Chan et al. (2021);

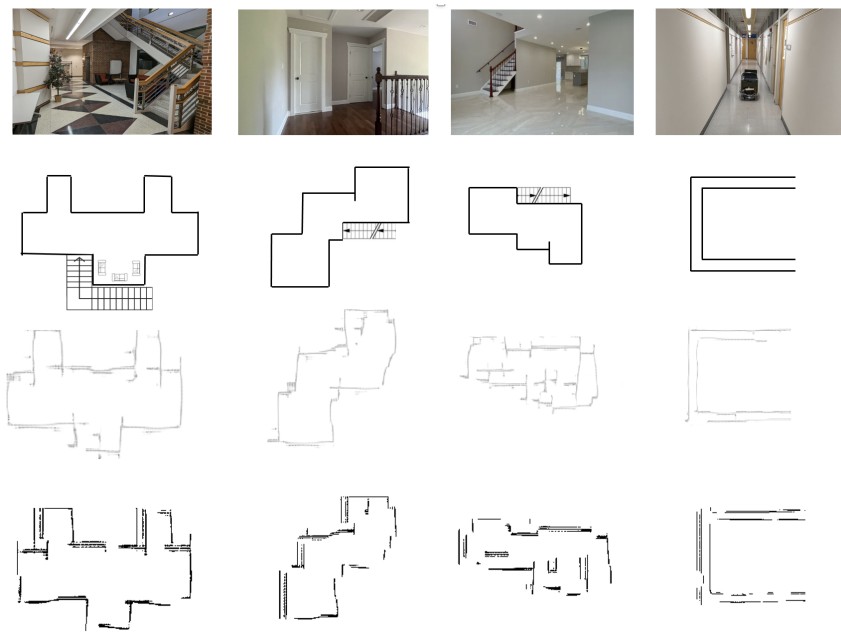

Figure 11: Real world results for `MapLearn`. Top row: indoor place image. Second row: ground-truth map. Third row: Hint-map's result. Fourth row: `MapLearn`'s result

Karam et al. (2020). As mentioned earlier, this incurs privacy concerns in places like homes Fraccaro et al. (2020). `MapLearn` alleviates this concern by essentially sensing a contour of the environment and filtering out the "high-frequency" details. Contour maps are helpful to many applications El-Sheimy & Li (2021). ■ **SLAM:** WiFi and audio SLAM Huang et al. (2011); Kreković et al. (2016) methods have also been popular, however, mapping in such scenarios refer to landmarks, i.e., localizing the WiFi access points or audio speaker locations Ferris et al. (2007); Liu et al. (2019). Imaging the contour of walls and large reflective objects entails different set of challenges. As an aside, Rhoomba robots perform visual SLAM, i.e., uses cameras to identify and localize landmarks in the home; naturally, research has reported privacy concerns through such vacuum cleaners Sami et al. (2020); Suryaprabha et al. (2022). ■ **RF Imaging and Synthetic Radar:** A rich body of work has developed algorithms for imaging the surroundings in the context of autonomous cars, or even in indoors, using drones or precisely moving robots that use special high-bandwidth transceivers Guan et al. (2020); Korany et al. (2019); Karanam & Mostofi (2017); Adib & Katabi (2013); Risbøl & Gustavsen (2018); Wang et al. (2017). Moreover, such RF techniques image a certain object in isolation, and get derailed when multipath reflections return from many other objects Zhao et al. (2018; 2019). For the problem of indoor mapping, RF reflections from adjacent rooms severely complicate this (inverse) problem Sen et al. (2013). `MapLearn`'s audio based approach naturally sidesteps that since audio signals get absorbed by walls. Moreover, using everyday mobile devices adds to the challenge. ■ **Audio based room shape estimation:** Mapping from audio has been investigated by Dokmanic's series of work where arrays of microphones are precisely calibrated; signal processing algorithms are proposed to solve inverse problems Dokmanić & Tashev (2014); Dokmanić et al. (2011); Kreković et al. (2016); Dokmanić et al. (2016). Lay home users are not capable of such precisely calibrated measurements in each room; walking around is much more feasible but the location errors render Dokmanic's approaches incompatible. `MapLearn` fills these practical gaps.

## 6    CONCLUSION

We find that a cGAN architecture, guided by spatial information from measured signals, can learn contour maps of indoor environments. The training of our model was performed on simulated data and yet the results from the real world was robust, generalizable. Our system `MapLearn` was evaluated on clutter-free environments; learning maps even in the presence of clutter is a natural next step. If successful, a solution can enable a host of societal applications.

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
