# OpenReview forum: "MapLearn: Indoor Mapping using Audio"
_ICLR.cc/2024/Conference — Submitted to ICLR 2024_

### Official Review · Reviewer_Cs1q · 2023-10-25

**Soundness:** 3 good
**Presentation:** 3 good
**Contribution:** 3 good
**Rating:** 5
**Confidence:** 4

**Summary:**

In this paper, a MapLearn method is proposed to generate indoor floorplan maps from audio signals by learning a conditional GAN. Instead of directly conditioning the GAN on the audio signals, the authors seek to derive spatial information from audio signals. Results on simulated and real-world data validate the effectiveness of MapLearn to some extent.

**Strengths:**

++ A spatial hint map derived from audio signals attains geometric information.

++ Using the envelope to mitigate the signal noise is insightful.

**Weaknesses:**

1) How is the envelope of h calculated?

2) What does "patch" mean for the patch GAN discriminator?

3) Could the authors add the precision and recall values of each estimated floor plan map to the third and fourth rows of Figure 10?

4) What if the MLP for corner prediction from the envelope is removed and directly using the envelope along with the hint map as the input?

5) Will the data and code be released for reproducibility?


Minor:
-- page 2: "it's introduction"

-- page 4: "a MLP"

-- page 5: "a MSE", "a L1"

-- Figure 5: "cGAN outpu"

**Questions:**

See the weaknesses above.

---

> ### Author Response · Authors · 2023-11-23
>
> #### Q1: How to calculate envelope h?
>
> Response: Thanks, we should have explained this better.
> We use Hilbert transform. We will make it clearer in the paper.
>
> #### Q2: What is PatchGAN?
>
> Response: We should have clarified this. PatchGAN is from the “Image-to-Image Translation with Conditional Adversarial Networks” paper in the reference. Instead of outputting a final real/fake value directly for the entire image, PatchGAN crops the image into patches and outputs a real/fake for each patch. A learnable function that fuses small patch results together is utilized for final real/fake decisions. This method has proven to be effective for recovering local details inside images.
>
> #### Q3: Add P(d) and R(d) for figure 10.
>
> Response:
> We report the P(40) and R(40) values for all the six floor plan visualizations here (we will add this table if the paper is accepted). The P and R values are all around 0.6, showing promising floor plan estimation result. For the full MapLearn pipeline, due to the fewer fake walls predicted, the P(40) value is overall better than that of HintMap only solution.
>
> |          | FP1         | FP2         | FP3         | FP4         | FP5         | FP6         |
> |----------|-------------|-------------|-------------|-------------|-------------|-------------|
> |          | P(40)/R(40) | P(40)/R(40) | P(40)/R(40) | P(40)/R(40) | P(40)/R(40) | P(40)/R(40) |
> | MapLearn | 0.522/0.497 | 0.720/0.559 | 0.472/0.484 | 0.533/0.504 | 0.655/0.566 | 0.689/0.592 |
> | HintMap  | 0.560/0.610 | 0.622/0.553 | 0.482/0.559 | 0.611/0.675 | 0.658/0.599 | 0.582/0.635 |
>
>
>
> #### Q4: Directly use the envelope along with the hint map as the input.
>
> Response: Thanks for the comment. The envelope should be used in conjunction with location information (i.e., the location where the measurement is being performed). We find that explicitly telling the network about measurement locations (and shifting the local map-patches to those locations) is beneficial. It is more generalizable than directly inputting the measurement location (x,y) as two other dimensions and relying on the neural network to learn.
>
> #### Q5: Will the data and code be released for reproducibility?
>
> Response: Yes, we will release the data and code if the paper is accepted.
>
>
> #### Q6: Minor typos
> Response: Thank you for identifying these issues. We will fix them.

---

### Official Review · Reviewer_qNkX · 2023-10-30

**Soundness:** 3 good
**Presentation:** 3 good
**Contribution:** 3 good
**Rating:** 6
**Confidence:** 4

**Summary:**

This paper tries to reconstruct the floorplan of an area via audio reflections. In detail, by accepting recorded audio reflections and corresponding locations as input, a conditional GAN is used to predict the floorplan contour as output. Experiments of the model trained on synthetic dataset gives promising accuracy on real datasets.

**Strengths:**

1.	Interesting idea. Cameras and Lidars are generally used to reconstruct the map of an environment. In the paper, the authors propose to use audio reflections to recover the floorplan of an environment. The idea is different to previous methods and sounds very interesting.

2.	Robust solution. Predicting the indoor map from only audio reflections is very challenging due to limited and sometimes noisy information in the audio reflections. Therefore, the authors introduce the Hint map to recover a coarse geometry of an area according to the time of receiving responses. The Hint map then is used to replace original signals as the input to cGAN, making results more accurate and robust.

3.	Individual room estimation. It is really difficult to recover the whole area all at once, so recovering one room individually and stitching rooms into floorplan with known locations should give better results.

**Weaknesses:**

1.	Limitation. My major concern is the limitation of the idea. Audio signals contain much less information than visual or lidar signals for map reconstruction, which means the floorplan generated from audio reflections are not that accurate as also shown in the experiments. Moreover, audio reflections vary in environments with different painting materials, decorations, and objects in the area. These factors may further impair the accuracy and generalization.

2.	Pose accuracy. In the paper, the location is obtained from IMU. However, IMU is very sensitive to error accumulation especially in large rooms. Currently, as GPS is not available in indoor environments, visual information is the best to obtain location information. My concern is that if we have visual information, why don’t we just recover an accurate map with SfM techniques from images? Note that audio reflections only provide very coarse floorplan.

3.	Applications of floorplan contour. In Section 1, the authors mention that ‘a simple floorplan contour may suffice in most cases’ but don’t provide examples. Maps reconstructed by cameras and Lidars can be used for localization which is the key technique to AR/VR and robotics navigation. It is not very clear to see how to use floorplan contour in real applications.

**Questions:**

Please see Weaknesses for details.

---

> ### Author Response · Authors · 2023-11-23
>
> #### Q1: Audio has low resolution leading to impaired mapping accuracy.
>
> Response: True, audio signals have much longer wavelengths compared to optical cameras and LIDARs, hence much lower spatial resolution.
> However, this lack of resolution hides rich details about the environment, leading to better privacy.
> We think many in-home applications (Alexa, digital twins, localization) may favor coarse-grained maps and strong privacy as opposed to the vice versa; in this tradeoff, audio lies in the sweet spot.
> Finally, even extracting this coarse-grained information is not easy; traditional signal processing methods do not work well.
> We have demonstrated that using cGAN can be effective.
>
> #### Q2: IMU location error will affect system performance.
>
> Response: Yes, this is a valid comment.
> We acknowledge the accumulation of IMU error would impact mapping.
> However, a growing body of literature is showing improved IMU-based localization [1,2] using (re)calibration opportunities from the environment (using Wifi, BLE, magnetic, and other sensory landmarks, etc. [3,4,5]).
> In sum, IMU-based drifts can be reset periodically  allowing the error to be bounded.
> We evaluated the sensitivity of map error as a function of IMU location error (Table 3).
> So long as the IMU error is unbiased, the mapping performance degrades gracefully with IMU error.
>
> #### Q3: Application of the MapLearn pipeline.
>
> Response: We mentioned the following to Reviewer 1 as well.
> Mapping with audio-reverberations is relevant when cameras or LIDARs are  not acceptable (e.g., privacy concerns in homes, poor lighting conditions, and unavailability of LIDARs).
> Potential use-cases include:
>
> + Digital twins of homes are useful for various applications, e.g., when away from home, a user wants to track her aging parent or her pet on a digital twin of the home floorplan.
> The floorplan is valuable for visualization.
> + Conversational AI agents (e.g., Alexa and Google Home) intend to understand the spatial context of a conversation with a user.
> Knowing the user's location is useful, but contextualizing that location on an even coarse-grained floorplan offers much more context (e.g., user is in kitchen, living room, etc.).
> Mapping with cameras/LIDARs raise privacy concerns inside homes.
> + Localization algorithms (e.g., particle filters) routinely use floorplans to update their posterior distributions. A coarse-grained floor plan with wall location uncertainties will still significantly improve trajectory estimation.
> + Floorplans are useful to signal processing applications, e.g., WiFi radios may beamform better with an understanding of walls around it.
> On similar lines, 3D surround sound can be synthesized better with knowledge of floorplans.
> + Firefighters may rapidly need to map out a floor plan or army troops may need to map out an enemy building; cameras may be inadequate due to smoke or poor lighting conditions.
> Audio reverberations (inaudible) should be more effective.
>
>
>
> ##### [1] Liu, Wenxin, et al. "Tlio: Tight learned inertial odometry." IEEE Robotics and Automation Letters 5.4 (2020): 5653-5660.
> ##### [2] Sun, Scott, Dennis Melamed, and Kris Kitani. "IDOL: Inertial deep orientation-estimation and localization." Proceedings of the AAAI Conference on Artificial Intelligence. Vol. 35. No. 7. 2021.
> ##### [3] Wang, He, et al. "No need to war-drive: Unsupervised indoor localization." Proceedings of the 10th international conference on Mobile systems, applications, and services. 2012.
> ##### [4] Chintalapudi, Krishna, Anand Padmanabha Iyer, and Venkata N. Padmanabhan. "Indoor localization without the pain." Proceedings of the sixteenth annual international conference on Mobile computing and networking. 2010.
> ##### [5] Shu, Yuanchao, et al. "Magicol: Indoor localization using pervasive magnetic field and opportunistic WiFi sensing." IEEE Journal on Selected Areas in Communications 33.7 (2015): 1443-1457.

---

### Official Review · Reviewer_ycoX · 2023-11-01

**Soundness:** 2 fair
**Presentation:** 3 good
**Contribution:** 3 good
**Rating:** 5
**Confidence:** 3

**Summary:**

The authors aim to generate floorplans of indoor environments from audio transmitted from a mobile device. They make a few assumptions –
1.	Known locations of the multiple walking paths where recordings were made.

2.	The rooms are devoid of clutter.

3.	The rooms are rectangular.

However, they do not assume complete coverage of the home and show results on 60% of the grid cells. The conditional GAN model (using a cGAN loss + certain audio processing priors) is pre-trained on simulated floor plans, and then tested on 4 real environments. The specific novel aspects of the architecture are –

1.	The paper suggests generating a hint_map based on the principles of signal propagation in reflective environment (an echo is a attenued time-delayed copy of the source signal). The authors use this fact to generate reflector circles corresponding to the peaks (impulse response) of the deconvolutions between the source and the echo.

2.	Instead of the raw impulse response, the paper suggests using the envelope of the response as that is less noisy.

3.	The impulse response is rotationally invariant, hence the paper inputs 4 measurements to the encoder to resolve this ambiguity.

**Strengths:**

1. The problem statement is novel and it appears that it has not been tackled earlier.

2. The authors propose exploiting audio and floorplan specific priors to generate floor plans. This is an interesting direction.

3. The end-to-end evaluation is reasonable.

**Weaknesses:**

- I'm not convinced about the motivation and practicality of this problem.
     - While I admit there are applications wherein privacy dictates that camera or LIDARs can't be used, wouldn't the same privacy concerns persist if an audio beacon is used?
     - What is the range of audio frequencies that this method can operate on (does it need to operate in the audible range)? I would imagine the application is expected to not operate in the audible range, please correct me on this if I'm wrong.

- Results in Section 4.1 is unclear. What does the HintMap baseline indicate? Isn't Hint map part of the MapLearn method? If so, what are the contribution of the rest of the components (ablations)?
- Question about the metrics and visualizations, the paper does not motivate why the improvement matters.
   - If an approximate floorplan suffices for downstream application(s), does the improvement in P(d) and R(d) matter?
   - It's difficult to interpret images in Fig 10, I'm not sure if Row 2 is better than Row 3 or vice versa.

- Demonstrating an application where a more accurate floorplan from audio is needed would be useful in showing the benefit of the improvement in floorplan estimate (as the goal is anyways to obtain an approximate floorplan).

**Questions:**

Please answer the questions listed in Weaknesses.

---

> ### Author Response · Authors · 2023-11-23
>
> #### Reviewer 2
> #### Q1: Motivation; why does acoustics not have privacy concerns?
>
> Response: Audio sensing will record ambient voices and sound events, which is indeed a privacy leakage.
> However, we have used the ultrasonic audio bands of 20kHz-40kHz (mentioned at the end of page 5).
> Human voice and music have no footprint in these frequencies.
> As a result, we can preserve privacy.
> Additionally, note that cameras end up ``seeing'' details of the environment (e.g., bathrooms, bedrooms, closets, text and images of documents on the table and fridge, etc.).
> Audio preserves this notion of privacy as well.
>
> #### Q2: What does hint map baseline indicate? How to interpret Fig 10 row 2 and 3?
>
> Response: The hint-map is the floorplan prior estimated from only the first audio reflection.
> The later reflections capture additional environmental information and contribute to boosting the paper's final performance.
> We have used the hint-map as a baseline (and as an ablation study to compare the value of first vs. late reflections).
> In  Figure 10, row 2 shows the ground truth and row 3 shows the estimation based on the hint-map alone.
> Row 4 shows the final results after including the late reflections.
> With hint maps, the estimated floor plan exhibits more fake walls because the neural network tries to insert walls at the boundary of open spaces. With the full MapLearn pipeline, fake walls are fewer.
>
>
> #### Q3: If approximate floorplan suffices, why does P(d), R(d) matter?
>
> Response:
> Thanks for recognizing this point.
> We have struggled to design reasonable metrics -- not too harsh, not too lenient -- that can capture the quality of learnt floorplans.
> Clearly, the application determines when an approximate floorplan is sufficient, but to develop some application-agnostic quantification, we decided on P(d) and R(d).
> This is by no means the perfect metric and we are very open to suggestions.
> We will continue to think about better metrics for this problem but believe that P(d), R(d) are reasonably fair measures of performance.
>
>
> #### Q4: Application where a more accurate floor plan from audio is needed.
>
> Response: When a user walks around a lot (covering say 80% of the home) then the hint-map generates a reasonably good floorplan (and the late audio reflections are less crucial).
> For such cases, the hint-map and MapLearn can both satisfy the same applications.
> However, when the user walks less, the hint-map degrades quicker and the value of late reflections becomes pronounced.
> If this paper is accepted, we will add results that visually compares floorplan accuracy between MapLearn and the hint-map, when the user has walked say 60%.
> This will highlight the additional accuracy gain from late reflections.
>
>
> Now, what applications will need more accuracy than just a good visual approximation?
> Many indoor localization applications use particle filters [1] on a floorplan.
> Very briefly, a user's possible location is modeled by many particles, each associated with a probability.
> Each particle is a hypothesis of where the user may be located.
> As the user moves, the particles are propagated in the measured walking direction.
> When a particle moves through a wall, these particles must be removed since they are an infeasible hypothesis.
> Said differently, the floorplan helps update the posterior distribution of the user's location.
> The floorplan accuracy directly impacts the convergence (and accuracy) in such Bayesian methods for localization.
>
>
> #### [1] Del Moral, Pierre. "Nonlinear filtering: Interacting particle resolution." Comptes Rendus de l'Académie des Sciences-Series I-Mathematics 325.6 (1997): 653-658.

---

### Official Review · Reviewer_R9N4 · 2023-11-06

**Soundness:** 2 fair
**Presentation:** 2 fair
**Contribution:** 2 fair
**Rating:** 3
**Confidence:** 3

**Summary:**

This paper proposes a map construction system by using acoustic signals. The problem is interesting and useful, and can support lots of location based services.

**Strengths:**

An interesting problem, to support location based services. The GAN architecture to improve robustness against acoustic measurement noises.

**Weaknesses:**

There are some recent work using smartphone's acoustic signals to measure the environment and construct the map, e.g., BatMapper, the only difference is the mode, thus please compare with some recent work.
I'm not sure how they train the GAN network, especially how they collect the ground truth. BatMapper doesn't need any ground truth. If they need the ground truth, it will harm its application and user experimence.
Their evaluations are based on simulation, and why not implement a prototype and conduct experiment in real buildings.

**Questions:**

1. How to gather the ground truth, and what is the usage scenario.
2. Comparison with recent work, e.g., BatMapper.
3. Develop a prototype and test in real buildings, instead of simulation.

---

> ### Author Response · Authors · 2023-11-23
>
> #### Q1: How to get ground-truth for training?
>
> Response: We trained our model entirely using simulated data from the ZinD 3D-scan dataset (cited and explained in Section 3). The ground-truth floorplans are part of this dataset.
> The real-world experiments were only performed during testing and results are promising (i.e., training on simulation data generalized to real environments).
> Of course, to report error for the real-world experiments, we obtained ground-truth floorplans from the facilities manager of the buildings.
>
>
> #### Q2: What is the usage scenario?
>
> Response: Mapping with audio-reverberations is relevant when cameras or LIDARs are  not acceptable (e.g., privacy concerns in homes, poor lighting conditions, presence of fog or smoke, and unavailability of LIDARs).
> Potential use-cases include:
>
> + Digital twins of homes are useful for various applications, e.g., when away from home, a user wants to track her aging parent or her pet on a digital twin of the home floorplan.
> The floorplan is valuable for visualization.
> + Conversational AI agents (e.g., Alexa and Google Home) intend to understand the spatial context of a conversation with a user.
> Knowing the user's location is useful, but contextualizing that location on a floorplan offers much more context (e.g., kitchen, living room, etc.).
> Mapping with cameras/LIDARs raise privacy concerns inside homes.
> + Localization algorithms or state estimation problems (e.g., particle filters) routinely use floorplans to update their posterior distributions.
> + Floorplans are useful to signal processing applications, e.g., WiFi radios may beamform better with an understanding of walls around it.
> On similar lines, 3D surround sound can be synthesized better with knowledge of floorplans.
> + Firefighters may rapidly need to map out a floor plan; cameras may be inadequate due to smoke and poor lighting conditions.
> Audio reverberations should be more effective.
>
>
>
> #### Q3: Comparison with BatMapper.
>
> Response: Thanks for bringing this up; we should have elaborated more in the paper.
> BatMapper uses a pure signal processing-based approach under the assumptions that: (1) floorplans are rectangular shaped rooms or corridors, and (2) users need to walk quite close to the walls.
> This is necessary because BatMapper only utilizes information from early reflections (similar to our hint map) to ultimately infer the parameters of the rectangle.
> Our model is crucially different since it uses all the audio reverberations (i.e., the room impulse response) to learn any layout of the environment (including partitions, L or U-shaped rooms, etc.).
> Results show that we need the user to walk less and our maps are far more complex than BatMapper (see our Figure 10 and 11 vs. BatMapper Figure 17).
>
>
> #### Q4: Develop a prototype and test in a real building.
>
> Response: We indeed implemented the prototype and performed the experiments in real world across 4 different buildings.
> Please find details regarding experiment setup in section 3 (Figure 8) and results in section 4 (Figure 11).
>
> #### Q5: The need for ground-truth harms its application and user experience.
>
> Response: We respectfully disagree with this.
> We only need ground-truth for training and since we train on simulation data, the ground-truth is easily available.
> We carefully performed the simulation (channel models developed on real signal propagation physics) so that it resembles real world as much as possible. This is why the proposed method works in real world as well.
> We do not need ground-truth when mapping an unknown floorplan in a real building (we validated this in 4 real buildings).

---

### Author Response · Authors · 2023-11-23

We thank reviewers for their thoughtful reviews. Please find our brief responses below.

---

### Meta-Review · Area_Chair_vbpX · 2023-12-06

**Metareview:**

The authors propose an algorithm for mapping indoor spaces (2D floorplans) using audio beacons (from a smartphone) and their recorded reflections. The architecture is based on cGAN, with additional constraints based on signal propagation. The idea is to generate a hint-map that captures the rough geometry based on first peak reflections, and room corner estimates computed from envelopes of RIRs as input to a cGAN that predicts the floor plan. The model is trained on simulated data, and is shown to generalize well to real, clutter-free homes.

The paper is a good example of using machine learning to solve a domain-specific problem, incorporating domain-specific knowledge. There are specific novelties like the use of hint_map based on signal propagation to constrain GAN, use of response envelope and ways to address rotational invariance of RIR. In general, the reviewers agreed that the use of audio and floorplan specific priors are interesting directions.

During the review, multiple reviewers questioned the practicality the proposed method is. There are several shortcomings, like the assumption about clutter free home and walking paths, practical applications of the presented approach over traditional signal processing approach that provide less accurate but reasonable floor plans (BatMapper), advantages over more accurate floorplan estimators that use Camera / LIDAR, etc. The authors argue about the privacy benefits, and some specific use cases like in the presence of smoke. They are interesting, but not too compelling. Over signal processing approaches, the method is more general and less constrained (the user need not walk around close to the wall). But this advantage seems somewhat limited.

There were also concerns about evaluations. The qualitative evaluations did not clearly show the benefits of the model over using the hint-map. The new quantitative metrics were also not easy to interpret (the authors were open to using new metrics, but unfortunately there was not clear suggestion on what a good metric would be). It would have als been useful to show some other baselines using techniques like BatMapper or using camera / LIDAR to show how they compare to MapLearn.

**Justification For Why Not Higher Score:**

The paper presents an interesting application of cGAN / machine learning to a domain specific problem. But as of now, while the ideas are interesting, the method needs to be developed more to be more robust and practical. The authors should also consider using better comparisons, both in terms of metrics and other approaches to show how well the method works.

**Justification For Why Not Lower Score:**

N/A

---

### Decision · Program_Chairs · 2024-01-16

Reject